# The Use of a Targeted Must Oxygenation Method in the Process of Developing the Archival Potential of Natural Wine

**Jozefína Pokrývková [1],*** , **Štefan Ailer [2]**, **Jaroslav Jedlička [2]**, **Peter Chlebo [3]** and **Ľuboš Jurík [1]**

[1] Department of Water Resources and Environmental Engineering (WREE), Slovak University of Agriculture in Nitra, 949 76 Nitra, Slovakia; lubos.jurik@uniag.sk

[2] Department of Fruit Science, Viticulture and Enology (DFSVE), Slovak University of Agriculture in Nitra, 949 76 Nitra, Slovakia; stefan.ailer@uniag.sk (Š.A.); jaroslav.jedlicka@uniag.sk (J.J.)

[3] Department of Human Nutrition (FAaFR), Slovak University of Agriculture in Nitra, 949 76 Nitra, Slovakia; peter.chlebo@uniag.sk

*   Correspondence: jozefina.pokryvkova@uniag.sk; Tel.: +42-19-0534-2489

**Abstract:** We examined the effect of two different technological processes for wine production on qualitative parameters of wine. We used the reductive method, which is currently considered to be the conventional method, and a targeted must oxidation method. We evaluated the basic physicochemical parameters and sensory attributes of wine as well as the content of phenolic substances in wine, which are responsible for the oxidation processes. The vegetable materials used were the grape varieties, Welschriesling, Chardonnay, and Rheinriesling. The content of phenolic substances was determined by HPLC (high-performance liquid chromatography), and the basic analytical parameters of wine were determined by FT-IR (Fourier Transform Infrared Spectroscopy) spectrometry. The sensory analysis was evaluated according to the International Union of Oenologists. For each of the wines examined, the total content of phenolic substances decreased after the targeted oxidation method was applied. For the Welschriesling variety produced by the reduction method, the total content of the 19 monitored phenolic substances in the year 2015 was $88.37 \, \text{mg} \cdot \text{L}^{-1}$, and for the wine produced by target oxygenation, it was $68.63 \, \text{mg} \cdot \text{L}^{-1}$. This represents a decrease of 21.5%. In the year 2016, the decrease was 20.91%. By reducing the content of phenolic substances, the oxidation processes in wines are eliminated after bottling. Thus, there is less need for sulphating wines with a reduced content of phenols.

**Keywords:** oxygen and grape must; phenolic substances; white wine ripening

## 1. Introduction

Phenolic substances are very unstable and highly reactive compounds that oxidize easily. They are a natural component of a component of plant tissue, in which they perform mainly a protective function against fungal diseases and growing stress [1]. A plant exposed to drought stress or salinization produces a higher amount of phenolic substances [2]. In addition to the protective function of plants, their phenolic substances have a high reactivity and antioxidant activity, which can be exploited in human treatments, such as the prevention of the cardiovascular diseases [3]. It is very important to know the properties of these substances in terms of their health effects, but also from a technological point of view, and to distinguish these effects with respect to color and archival potential of wine. Except for the high reactivity, phenolic substances, especially caffeic acid, form aggregates with wine proteins, which, at a certain stage, may become insoluble and thus cause wine turbidity [4]. The effect of phenolic

substances on the color of wine is extremely significant. The oxidation of some phenols produces colored unsaturated conjugated ketones (quinones), which are the precursors to wine browning [5].

By gently processing the grapes, the content of the phenolic substances in white wine ranges by up to 250 mg·L$^{-1}$. In red wine, its content can be up to 4500 mg·L$^{-1}$ [6]. Phenolic concentration varies according to the wine production method and the wine variety used. Some phenolic substances that bind to sugar in must are released only during wine fermentation and maturation [7,8]. According to their chemical structure, we can divide phenolic substances into several groups [9]. Many authors divide phenolic substances into two big groups:

1.   substances of a flavonoid nature (anthocyanins, flavan-3ols, flavanols, and dihydroflavonols); and
2.   substances of a non-flavonoid nature (hydroxybenzoic and hydroxycinnamic acids, stilbenes, and volatile phenols).

Phenolic substances are those which remain in wine until they finally form its color. They lead to the browning of wine. Browning modifies the green and the yellow colors to brown. The higher the degree of oxidation is, the more intense is the browning [6]. They add phenolic substances to young wines, as they contain free sulfur dioxide, which can improve the structure (sensory fullness) of the wine [7]. Later, they are also harmful in white wine, because they catalyze the decomposition of the fruit aromas, creating an "old-fashioned aroma and taste" and sometimes wine browning. Gently processing the grapes and eliminating the phenolic content before the start of the alcoholic fermentation is one of the most important aspects of the natural prevention of oxidation and browning of wine [10].

By removing potentially oxidizable substances from the must oxygenating them, we tried to achieve a higher maturation and archiving potential of wine. The oxidation of must naturally takes place enzymatically, it is much more vigorous than in the case of wine, and the only by-product is water. The oxidation of wine partially takes place enzymatically, but it is mainly a chemical process. This process produces peroxides, which have adverse effects on the quality of wine.

The impact of the variability of climate on agriculture is never more obvious than in the case of viticulture and wine production, where climate is probably the most important aspect of grape ripening with respect to achieving the optimal characteristics for the production of a given style of wine [11].

One of the aims of the present research was to evaluate the influence of the targeted oxygenation of must on the sensory parameters of wine. The aim of the research was to monitor two different technological processes of grape processing relating to the elimination of the oxidation processes in wine and the management of its archiving potential while maintaining its quality parameters. In the experiments, we used two methods: a reductive method for processing grapes (control variant) and a method involving the targeted oxygenation of must.

## 2. Materials and Methods

Using the two abovementioned methods, we evaluated the basic physicochemical parameters, the content of phenolic substances, which are responsible for the oxidative processes in wine, and the sensory descriptions of wine. The sensory analysis is the most important in wine technology. The conditions of the sensory analysis were governed by a standard [12]. We statistically evaluated the content of the investigated phenolic substances.

### 2.1. Vegetable Materials

Grapes of Chardonnay and Riesling varieties came from the Modra locality in the Little Carpathian wine-growing region. Grapes of the Welschriesling variety came from the Farná locality from the South Slovak wine-growing region.

#### 2.1.1. Chardonnay

Chardonnay became an integral part of our assortment of grapes from the production of high-quality wines. The fullness and the harmony of wines and their typical grape variety bouquet

are peculiar to them. It is a kind of vivid grape variety, which is testified by the quality of the wine all over the world. At our latitudes, it is even accentuated by its aromatic substances. The aroma is distinctive and flowery, and the wines are delicate, elegant, and really harmonious [13]. The vineyard was 8 years old.

### 2.1.2. Rheinriesling

Rheinriesling is especially valued from the processing point of view. It provides an excellent bouquet and peppery and flowery notes, and it is as smooth as velvet, which reminds one of the smell of a lime tree. It has a significant content of delicate wine acid, which, together with the other ingredients, enables it to preserve the high quality of wines after many years of ripening [13]. The vineyard was 18 years old.

### 2.1.3. Welschriesling

Welschriesling is a very reliable grape variety, and it is suitable for the wine production of all quality categories. The wines have a distinctive, much gentler aroma profile than those of Rheinriesling. Welschriesling has a sufficient content of acids, which enables it to preserve the same quality of the wines, even in the case of an older vintage. It produces a naturally sweet wine [13]. The vineyard was 11 years old.

The content of sugar in the individual grape varieties at the time of grape harvesting ranged from 210 to 235 g·L$^{-1}$. We harvested the grapes by hand in both years in the second and the third weeks of September. For both the control and the experimental variants, must from the same grape was used for each variety.

Each variable was produced as a triple batch of 1.000 L. Three measurements were performed with each batch for verification. The article presents the average of the values from the measurements.

### 2.2. The Territorial Characteristics

We carried out research in the wine-growing years, 2015 and 2016. The years 2015 and 2016 were partially influenced by the climate cycle, El Niňo, but the others were not influenced by the warm phase of El Niňo, as the greenhouse effect contributes to global warming much more. In addition, the research was carried out during a period with the least sunshine since 1940 [14].

In the year 2015, there was a remarkably large number of days without any precipitation, and precipitation occurred mostly during unique stormy downpours and showers. In Hurbanovo, for example, a local rainfall in one day of 90.2 mm was recorded, and the average summer precipitation was 140.2 mm (Table 1).

**Table 1.** The average rainfall and temperatures in Hurbanovo in 2015 and 2016 during the growing season.

| Year | Average Amount of Precipitation During the Growing Season | Maximum Rainfall for One Day in Summer | Average Temperature during the Growing Season | Number of Super Tropical Days | Maximum Daily Temperature during Summer |
|------|------|------|------|------|------|
| 2015 | 140.2 mm | 90.2 mm | 22.9 °C | 22 | 35 °C |
| 2016 | 101.5 mm | 81.2 mm | 23.8 °C | 18 | 36.2 °C |

The temperature of the air was, on average, 1−1.5 °C higher than it was supposed to be on 1 July, and it was even about 2−3 °C higher than the model estimations of the world centers. The total amount of precipitation was substantially lower, but the assumption of big territorial differences in torrential precipitation proved to be accurate.

The average summer temperature in 2015 in Hurbanovo was 22.9 °C, and it was one of the warmest summers from the beginning of the observation, with 22 subtropical days and a maximum temperature of 35 °C or even higher. The year 2016 was considered very or exceptionally warm

compared with what was normal for the climate from 1961–1990. The average territorial divergence from the normal for the years 1961–1990 was +1.4 °C (from +1.1 to +1.8 °C). The year 2016 was ranked as the 7th warmest year since 1931, and in Hurbanovo, it was ranked the 8th warmest year since 1901. It is interesting that, in the year 2016, there were about 2 to 7 more summer days than during the extremely warm year of 2015.

On the other hand, in the year 2016, there was a considerably lower number of tropical days than in the previous year (approximately 15−30 fewer days). This may be because 2016 was rich in atmospheric precipitation. According to a preliminary evaluation, the total annual territorial precipitation of the last year ranked among the years with the highest rainfall since 1881.

The average precipitation in the territory of Slovakia was 895 mm, which represents 122% of the normal for 1961−1990. In February, it had the highest amount of precipitation (320% of the normal), followed by October (218% of the normal) and July (191% of the normal). The lowest amount of precipitation occurred in December (57% of the normal), in June (65% of the normal), and in March (66% of the normal) [15].

### 2.3. Technological Processes

In the control variant, after grinding the grapes, we added 50 mL·hL$^{-1}$ of dry ice, 15 mL·hL$^{-1}$ of pectolytic enzymes, and 35 mg·L$^{-1}$ of Sulphur dioxide to the must in the batches, as recommended by the manufacturer. We pressed the must with a pneumatic press. The maximum pressure used was 0.15 MPa. The obtained must was statically decanted with bentonite for 24 hours.

In the experiment of the targeted oxygenation technology, we added 50 mL· hL$^{-1}$ of dry ice and 15 mL·hL$^{-1}$ of pectolytic enzymes to the must in the batches, as recommended by the manufacturer. We pressed the must with a pneumatic press. We did not use Sulphur dioxide or any other antioxidants in this variant until the must desilting phase. We removed the sludge by gravity using sodium-calcium bentonite at a dose of 60 g·hL$^{-1}$. The sludge removal took 16 hours. Oxygen was not introduced into the wine under atmospheric air pressure. The acidification took place naturally, in that no sulfur dioxide was added to the must until the desludging phase, thus the phenolic substances could react with the naturally absorbed atmospheric oxygen. The total time of the targeted oxygenation was 24 hours. We removed some of the phenolic substances oxidized by atmospheric oxygen from the must after sedimentation with sludge. Another technological process of wine production was identical for both variants. Fermentation needs a pure wine yeast culture, and fermentation requires that the temperature is in the range of 16−21 °C. The commercial phylum of delicate active, dry, wine yeast of *Saccharomyces cerevisiae* was used at a dose of 15 g·hL$^{-1}$. Fermentation was realized at a temperature of 14−15 °C (± 1.0 °C). Fermentation lasted 12 days. After fermenting, the wine was kept on yeast lees, without air admittance, for two weeks.

### 2.4. Chromatographic Method

Before determining the individual parameters, we excluded the wines by increasing the possible vacuum filtration and centrifuging (3000× *g*; 6 min). For all wine samples, we determined the values of the basic analytical parameters: ethanol, glucose, fructose, acid composition, total acids, pH value glycerol, and density. We used the FT-IR (Fourier Transform Infrared Spectroscopy) spectrometry method with the ATR (Attenuated total reflection) measurement procedure.

An analytical determination of the monitored phenolic substances was performed by liquid chromatography (HPLC). It is a separation and, at the same time, an analytical physicochemical method for the separation and the analysis of a mixture of substances, the basic principle of which is the division of the components of a mixture into mobile and stationary phases [16].

The analytical determination of the monitored phenolic substances was conducted using liquid chromatography (HPLC).

A mobile phase in this case is liquid. The fixed phase is a film of the corresponding/relevant substance settled on a bearer surface or fixed/stable adsorbent. The device, realizing HPLC

analyses, is called the liquid chromatograph [17]. We used standard solutions of acetonitrile (ACN) and methanol (MeOH), and they served as supergradients of purity. Katechen, epikatechen, vanil acid, protokatech acid, 4-hydroxybenzoo acid, gall acid, syring acid, p-kumar acid, trans-resveratrol, coffee acid, ferul acid, piceatannol, rutin, myricetin, quercetin, kaempferol, izorhamnetin, p-dimethylaminocinnamaldehyd (DMACA), Folin–Ciocalteu agent, 2.2-difenyl-β-pikrylhydrazyl radical (DPPH), 2.4.6-tripyridyl-s-triazin (TPTZ), and perchloric acid were originally from Sigma Chemical Co. (St. Louis, MO). Malvidin–3.5-diglukozid was from Indofine Chemical Company Inc. (Hillsborough, NJ).

*2.5. Sensorial Wine Evaluation Method*

We evaluated the organoleptic properties with a 100 point system according to the International Union of Oenologists. We sensorially evaluated the wines twice [18]. We performed two evaluations over a period of 10 months to compare the archival potential of wines produced by both technologies. The first evaluation had the task of sensorially evaluating young wines that were ready to be bottled during that period. In the second evaluation, we also assessed the archival potential of wines. For each sample, in addition to sensory evaluation, we also assessed the aromatic and the taste profiles of wines with specific descriptors.

The evaluation of the organoleptic quality was realized by an updated 100 point system, according to the international oenology union [18]. The wine was evaluated by 6 evaluators with relevant certificates and experience, and the extreme values were eliminated before fixing the arithmetic mean. From the point of view of the sensorial evaluation, appearance (purity and color), aroma (positive intensity of aroma, purity, and aroma harmony), and taste (positive intensity, purity, harmony, and persistence) were evaluated. The last evaluated parameter was the total impression of the wine. Each sample was not only judged by the sensorial evaluation but also by the specific descriptors from the point of view of the aromatic and the taste profiles of the wine.

From the point of view of the aromatic profile, the following descriptors were evaluated: ripe grape, blossoming meadow, healthy hay, apple, pepper, almond, walnut, vanilla, minerality/quartz, copper, tropical fruit, chlorophyll, vegetation, oxidation, yeast decomposition, others/error.

To ensure the same conditions for the aging of the wine, the samples were stored in the same storage areas at a temperature of 17–20 °C.

*2.6. Statistic Evaluation*

For the statistical evaluation of the results, we used the method of analysis of variance in the program, Statgraphics Centurion XVII (StatPoint Inc., Virginia, USA, 2016, version 17.01.03). We evaluated the influence of the variant and the year on the content of the monitored substances in the wine by a one-factor analysis of the variance of the dependent variable. We used an LSD test (least significant difference test, $P \leq 0.05$) to test the statistical significance of the results. This test is used to test a precisely determined number of hypotheses.

## 3. Results

The results of the basic physicochemical analyses of the examined wines are presented in Tables 2 and 3. The selected technology does not have a significant effect on the content of the basic physicochemical parameters in wine. For the wine from the Welschriesling and the Rheinriesling varieties, in both monitored years, we found lower contents of free and total $SO_2$ in variants that were processed by targeted oxygenation, while that fact did not negatively affect the sensory profiles of the wine. It was confirmed that, with the targeted oxidation technology, it is possible to spread wines more economically. The content of total Sulphur dioxide in the variants with targeted oxygenation was from 11–16% lower than in the reductive variants. These values were measured, after the clarification of the wine, with technological auxiliaries. For the Chardonnay variety from 2015, using both variants, it was necessary to use twice the high dose of $SO_2$ in the production plant when bottling the wine for

technological reasons. Therefore, it is not possible to evaluate the monitored parameter exactly for such wine (Table 2).

**Table 2.** Basic physicochemical parameters of wine from individual variants in 2015 with the standard deviations.

| Variant/ Parameter | $mg \cdot L^{-1}$ | $mg \cdot L^{-1}$ | % vol. | $g \cdot L^{-1}$ | - | $g \cdot L^{-1}$ | $g \cdot L^{-1}$ | $g \cdot L^{-1}$ | - |
|---|---|---|---|---|---|---|---|---|---|
| | SO$_2$ Free | SO$_2$ Total | Ethanol | Total Acid | pH | Glycerol | Glucose | Fructose | Density |
| W control variant | 17 ± 1 | 71 ± 1 | 12.32 ± 0.1 | 6.16 ± 0.03 | 3.12 ± 0.01 | 6.68 ± 0.01 | 0.00 ± 0.0 | 0.94 ± 0.11 | 0.991199 ± 0.02 |
| W oxygenation | 6 ± 1 | 47 ± 1 | 11.86 ± 0.1 | 6.04 ± 0.2 | 3.13 ± 0.00 | 7.12 ± 0.03 | 0.06 ± 0.01 | 0.83 ± 0.0 | 0.99187 ± 0.011 |
| RIE control variant | 11 ± 1 | 71 ± 2 | 13.60 ± 0.2 | 7.42 ± 0.11 | 3.15 ± 0.01 | 6.97 ± 0.0 | 4.77 ± 0.16 | 5.71 ± 0.04 | 0.99425 ± 0.01 |
| RIE oxygenation | 10 ± 1 | 57 ± 1 | 13.65 ± 0.1 | 7.08 ± 0.04 | 3.17 ± 0.00 | 7.38 ± 0.01 | 3.83 ± 0.18 | 4.09 ± 0.099 | 0.99351 ± 0.02 |
| CH control variant | 27 ± 1 | 69 ± 1 | 12.07 ± 0.1 | 7.34 ± 0.08 | 3.15 ± 0.00 | 7.14 ± 0.01 | 0.05 ± 0.0 | 1.03 ± 0.015 | 0.99280 ± 0.01 |
| CH oxygenation | 58 ± 2 | 136 ± 2 | 12.18 ± 0.2 | 7.55 ± 0.02 | 3.07 ± 0.01 | 7.47 ± 0.02 | 0.27 ± 0.01 | 0.67 ± 0.0 | 0.99289 ± 0.01 |

Legend: W – Welschriesling; RIE – Rheinriesling; CH – Chardonnay.

**Table 3.** Basic physicochemical parameters of wine from individual variants in 2016 with the standard deviations.

| Variant/ Parameter | $mg \cdot L^{-1}$ | $mg \cdot L^{-1}$ | % vol. | $g \cdot L^{-1}$ | - | $g \cdot L^{-1}$ | $g \cdot L^{-1}$ | $g \cdot L^{-1}$ | - |
|---|---|---|---|---|---|---|---|---|---|
| | SO$_2$ Free | SO$_2$ Total | Ethanol | Total Acid | pH | Glycerol | Glucose | Fructose | Density |
| W control variant | 25 ± 2 | 82 ± 1 | 12.49 ± 0.2 | 5.87 ± 0.02 | 3.27 ± 0.01 | 7.55 ± 0.02 | 1.27 ± 0.01 | 0.57 ± 0.00 | 0.9900 ± 0.01 |
| W oxygenation | 14 ± 1 | 68 ± 1 | 12.28 ± 0.3 | 5.27 ± 0.11 | 3.35 ± 0.02 | 6.98 ± 0.00 | 1.06 ± 0.012 | 0.44 ± 0.01 | 0.9947 ± 0.02 |
| RIE control variant | 18 ± 1 | 82 ± 2 | 13.15 ± 0.1 | 7.58 ± 0.03 | 3.21 ± 0.00 | 6.97 ± 0.01 | 1.37 ± 0.01 | 1.91 ± 0.04 | 0.99645 ± 0.011 |
| RIE oxygenation | 14 ± 1 | 67 ± 3 | 13.22 ± 0.1 | 7.47 ± 0.04 | 3.23 ± 0.01 | 6.38 ± 0.00 | 1.23 ± 0.014 | 1.08 ± 0.02 | 0.99721 ± 0.01 |
| CH control variant | 35 ± 3 | 78 ± 1 | 12.20 ± 0.2 | 7.82 ± 0.01 | 3.27 ± 0.00 | 6.14 ± 0.02 | 0.85 ± 0.00 | 1.23 ± 0.00 | 0.99630 ± 0.02 |
| CH oxygenation | 29 ± 2 | 65 ± 2 | 12.27 ± 0.2 | 7.74 ± 0.01 | 3.21 ± 0.00 | 6.27 ± 0.01 | 0.77 ± 0.00 | 1.17 ± 0.01 | 0.99699 ± 0.01 |

Legend: W – Welschriesling; RIE – Rheinriesling; CH – Chardonnay.

The present research was originally concerned with phenolic compounds. Many authors examine the phenolic substances in wine in terms of their health benefits and also their harmfulness in terms of wine aging. However, the direct effect of the oxidation of must on their content in wine is very insignificant. After the use of the targeted oxidation method, the total content of 19 monitored phenolic substances in each of the examined wines decreased. This decrease was the most significant in the Riesling variety. The results are shown in Tables 4–7, and the statistical evaluation is shown in Tables 8–10 and Figures 1–3.

**Table 4.** Content of 10 monitored phenolic substances in mg·L$^{-1}$ from individual variants in 2015 with the standard deviations.

| | | | Phenolic Substance | | | |
|---|---|---|---|---|---|---|
| Variant | W Control Variant | W Oxygenation | RIE Control Variant | RIE Oxygenation | CH Control Variant | CH Oxygenation |
| Galic Acid | 17.952 ± 0.75 | 15.996 ± 0.75 | 2.612 ± 0.50 | 2.603 ± 0.38 | 2.125 ± 0.56 | 1.852 ± 0.56 |
| Protocatechic Acid | 0.730 ± 0.01 | 0.415 ± 0.02 | 0.500 ± 0.01 | 0.567 ± 0.09 | 1.510 ± 0.025 | 1.148 ± 0.038 |
| Vanilic Acid | 0.328 ± 0.02 | 0.275 ± 0.018 | 0.503 ± 0.03 | 0.560 ± 0.021 | 0.524 ± 0.016 | 0.481 ± 0.012 |
| Galic Acid Ethylester | 3.191 ± 0.096 | 2.796 ± 0.007 | 0.321 ± 0.012 | 0.284 ± 0.003 | 0.555 ± 0.013 | 0.505 ± 0.015 |
| Total Caffeic Acid | 38.798 ± 0.98 | 32.647 ± 0.51 | 48.145 ± 0.39 | 46.052 ± 0.52 | 38.007 ± 0.95 | 33.918 ± 0.59 |
| Total Kumaron Acid | 9.932 ± 0.03 | 6.948 ± 0.2 | 5.481 ± 0.02 | 5.039 ± 0.3 | 6.870 ± 0.02 | 4.878 ± 0.14 |
| Total Ferulic Acid | 3.447 ± 0.155 | 3.140 ± 0.085 | 6.240 ± 0.074 | 5.717 ± 0.082 | 5.819 ± 0.025 | 5.555 ± 0.081 |
| Catechin | 6.321 ± 0.35 | 3.078 ± 0.23 | 2.466 ± 0.35 | 2.699 ± 0.24 | 7.495 ± 0.52 | 5.439 ± 0.52 |
| Epicatechin | 2.788 ± 0.01 | 1.786 ± 0.0 | 1.262 ± 0.0 | 1.268 ± 0.02 | 2.600 ± 0.0 | 2.022 ± 0.01 |
| Rutin | 0.960 ± 0.01 | 0.190 ± 0.02 | 0.407 ± 0.0 | 0.649 ± 0.01 | 2.833 ± 0.02 | 2.185 ± 0.0 |

**Table 5.** Content of nine other monitored phenolic substances in mg·L$^{-1}$ from individual variants in 2015. The total amount of all 19 monitored substances and their percentage reductions from the variants that were processed by oxygenation in comparison with the control variant with the standard deviations.

| | Phenolic Substance | | | | | |
|---|---|---|---|---|---|---|
| Variant | W Control Variant | W Oxygenation | RIE Control Variant | RIE Oxygenation | CH Control Variant | CH Oxygenation |
| 4-Hydroxibenzoic acid | 0.317 ± 0.0 | 0.276 ± 0.0 | 0.335 ± 0.01 | 0.288 ± 0.0 | 0.884 ± 0.02 | 0.512 ± 0.01 |
| Syringic acid | 0.383 ± 0.01 | 0.228 ± 0.0 | 0.246 ± 0.01 | 0.249 ± 0.0 | 0.404 ± 0.01 | 0.378 ± 0.01 |
| Trans-resveratrol-lin | 0.210 ± 0.01 | 0.077 ± 0.09 | 0.096 ± 0.00 | 0.103 ± 0.01 | 0.195 ± 0.01 | 0.129 ± 0.00 |
| Trans-piceid (free resveratrol) | 0.298 ± 0.00 | 0.132 ± 0.00 | 0.150 ± 0.01 | 0.118 ± 0.00 | 0.307 ± 0.013 | 0.133 ± 0.00 |
| Cis-resveratrol | 0.450 ± 0.02 | 0.096 ± 0.00 | 0.253 ± 0.012 | 0.270 ± 0.015 | 0.277 ± 0.01 | 0.269 ± 0.011 |
| Cis-piceid (free resveratrol) | 0.901 ± 0.00 | 0.392 ± 0.011 | 0.489 ± 0.013 | 0.462 ± 0.010 | 0.610 ± 0.014 | 0.480 ± 0.011 |
| Trans-piceatannol | 0.002 ± 0.00 | 0.002 ± 0.00 | 0.006 ± 0.00 | 0.010 ± 0.01 | 0.054 ± 0.00 | 0.038 ± 0.01 |
| Quercitrin - ß-D-Glukosid | 0.351 ± 0.00 | 0.133 ± 0.00 | 0.087 ± 0.00 | 0.092 ± 0.00 | 0.114 ± 0.01 | 0.086 ± 0.00 |
| Quercitrin | 0.011 ± 0.0 | 0.019 ± 0.0 | 0.009 ± 0.0 | 0.018 ± 0.0 | 0.035 ± 0.0 | 0.038 ± 0.0 |
| Total (mg·L$^{-1}$) | **87.37 ± 1.25** | **68.63 ± 0.85** | **69.61 ± 0.92** | **67.05 ± 0.89** | **71.22 ± 1.11** | **60.04 ± 0.98** |
| Reduction (%) | | *21.5* | | *3.7* | | *15.7* |

**Table 6.** Content of 10 monitored substances in mg·L$^{-1}$ from individual variants in 2016 with the standard deviations.

| | Phenolic Substance | | | | | |
|---|---|---|---|---|---|---|
| Variant | W Control Variant | W Oxygenation | RIE Control Variant | RIE Oxygenation | CH Control Variant | CH Oxygenation |
| Galic Acid | 3.174 ± 0.25 | 1.719 ± 0.3 | 1.712 ± 0.38 | 1.403 ± 0.21 | 2.021 ± 0.35 | 1.742 ± 0.25 |
| Protocatechic Acid | 0.518 ± 0.02 | 0.593 ± 0.01 | 0.520 ± 0.01 | 0.548 ± 0.012 | 1.414 ± 0.01 | 1.182 ± 0.01 |
| Vanilic acid | 0.093 ± 0.013 | 0.108 ± 0.015 | 0.443 ± 0.02 | 0.430 ± 0.01 | 0.426 ± 0.03 | 0.410 ± 0.02 |
| Galic Acid Ethylester | 0.262 ± 0.015 | 0.140 ± 0.018 | 0.281 ± 0.013 | 0.263 ± 0.017 | 0.524 ± 0.01 | 0.488 ± 0.01 |
| Total Caffeic Acid | 47.229 ± 0.16 | 37.013 ± 0.11 | 46.344 ± 0.02 | 44.432 ± 0.03 | 35.217 ± 0.02 | 32.847 ± 0.11 |
| Total Kumaron Acid | 13.610 ± 0.4 | 11.476 ± 0.2 | 5.147 ± 0.1 | 4.752 ± 0.1 | 6.421 ± 0.2 | 4.498 ± 0.3 |
| Total Ferulic Acid | 2.826 ± 0.062 | 2.216 ± 0.025 | 5.840 ± 0.018 | 5.414 ± 0.029 | 5.623 ± 0.075 | 5.329 ± 0.081 |
| Catechin | 4.157 ± 0.35 | 3.629 ± 0.28 | 2.321 ± 0.45 | 2.178 ± 0.25 | 6.782 ± 0.52 | 4.578 ± 0.32 |
| Epicatechin | 1.526 ± 0.01 | 1.214 ± 0.0 | 1.078 ± 0.0 | 1.019 ± 0.0 | 2.358 ± 0.01 | 1.891 ± 0.0 |
| Rutin | 0.298 ± 0.0 | 0.382 ± 0.02 | 0.307 ± 0.01 | 0.327 ± 0.0 | 2.789 ± 0.0 | 2.577 ± 0.01 |

**Table 7.** Content of nine other monitored phenolic substances in mg·L$^{-1}$ from individual variants in 2016. The total amount of all 19 monitored substances and their percentage reductions from the variants processed by oxygenation in comparison with the control variant with the standard deviations.

| | Phenolic Substance | | | | | |
|---|---|---|---|---|---|---|
| Variant | W Control Variant | W Oxygenation | RIE Control Variant | RIE Oxygenation | CH Control Variant | CH Oxygenation |
| 4-Hydroxibenzoic acid | 0.093 ± 0.001 | 0.108 ± 0.00 | 0.315 ± 0.00 | 0.288 ± 0.01 | 0.841 ± 0.01 | 0.784 ± 0.02 |
| Syringic acid | 0.242 ± 0.02 | 0.208 ± 0.012 | 0.346 ± 0.01 | 0.287 ± 0.00 | 0.377 ± 0.018 | 0.319 ± 0.00 |
| Trans-resveratrol-lin | 0.066 ± 0.01 | 0.034 ± 0.00 | 0.127 ± 0.00 | 0.109 ± 0.00 | 0.214 ± 0.01 | 0.207 ± 0.01 |
| Trans-piceid (free resveratrol) | 0.214 ± 0.025 | 0.168 ± 0.018 | 0.134 ± 0.013 | 0.117 ± 0.00 | 0.312 ± 0.00 | 0.198 ± 0.021 |
| Cis-resveratrol | 0.377 ± 0.00 | 0.281 ± 0.01 | 0.284 ± 0.00 | 0.291 ± 0.01 | 0.287 ± 0.00 | 0.274 ± 0.00 |
| Cis-piceid (free resveratrol) | 0.214 ± 0.01 | 0.168 ± 0.00 | 0.417 ± 0.02 | 0.409 ± 0.01 | 0.574 ± 0.00 | 0.397 ± 0.02 |
| Trans-piceatannol | 0.009 ± 0.00 | 0.004 ± 0.00 | 0.006 ± 0.00 | 0.017 ± 0.00 | 0.047 ± 0.00 | 0.038 ± 0.00 |
| Quercitrin - ß-D-Glukosid | 0.019 ± 0.00 | 0.070 ± 0.00 | 0.074 ± 0.00 | 0.063 ± 0.00 | 0.098 ± 0.00 | 0.071 ± 0.00 |
| Quercitrin | 0.020 ± 0.00 | 0.019 ± 0.00 | 0.007 ± 0.00 | 0.011 ± 0.00 | 0.023 ± 0.00 | 0.031 ± 0.00 |
| Total (mg·L$^{-1}$) | **72.03 ± 0.98** | **56.96 ± 0.89** | **65.703 ± 0.78** | **62.358 ± 0.97** | **66.348 ± 0.76** | **57.861 ± 1.12** |
| Reduction (%) | | *20.91* | | *5.1* | | *12.8* |

**Table 8.** Influence of the variety and the variant on the total content of 19 investigated phenolic substances.

| Method: 95.0 Percent LSD | | | |
|---|---|---|---|
| **Variant** | **Repetition** | **LS Average** | **Homogenous Group** |
| CHO | 3 | 61.5438 | X |
| RIEO | 3 | 68.5479 | XX |
| WO | 3 | 70.1262 | X |
| RIEK | 3 | 71.1078 | X |
| CHK | 3 | 72.7154 | X |
| WK | 3 | 88.8682 | X |

CHO - Chardonnay oxygenation, RIEO - Rheinriesling oxygenation, WO - Welschriesling oxygenation, CHK - Chardonnay control variant, RIEK - Rheinriesling control variant, WK - Welschriesling control variant.

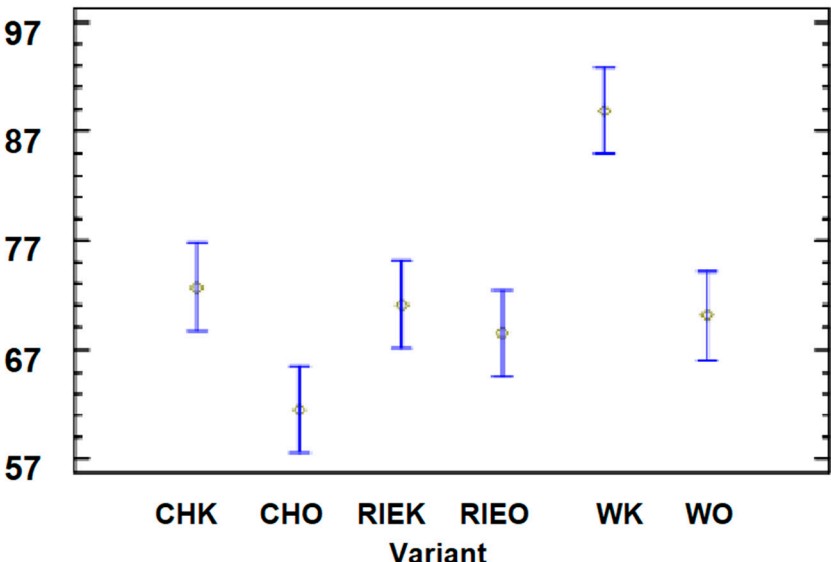

**Figure 1.** Graphic representation of the 95% confidence intervals of the sum of 19 monitored phenolic substances for the tested averages from individual varieties and variants in 2015.

**Table 9.** Influence of the variant on the total content of 19 monitored phenolic substances in wine in 2015 and 2016.

| Method: 95.0 Percent LSD | | | |
|---|---|---|---|
| **Variant** | **Repetition** | **LS Average** | **Homogenous Group** |
| WO | 6 | 63.5452 | X |
| WK | 6 | 80.4471 | X |

**Table 10.** Influence of the year on the total content of 19 monitored phenolic substances in wine in 2015 and 2016 for the RIE varieties.

| Method: 95.0 Percent LSD (Least Significant Difference) | | | |
|---|---|---|---|
| **Year** | **Repetition** | **LS Average** | **Homogenous Group** |
| 2016 | 6 | 64.495 | X |
| 2015 | 6 | 79.4972 | X |

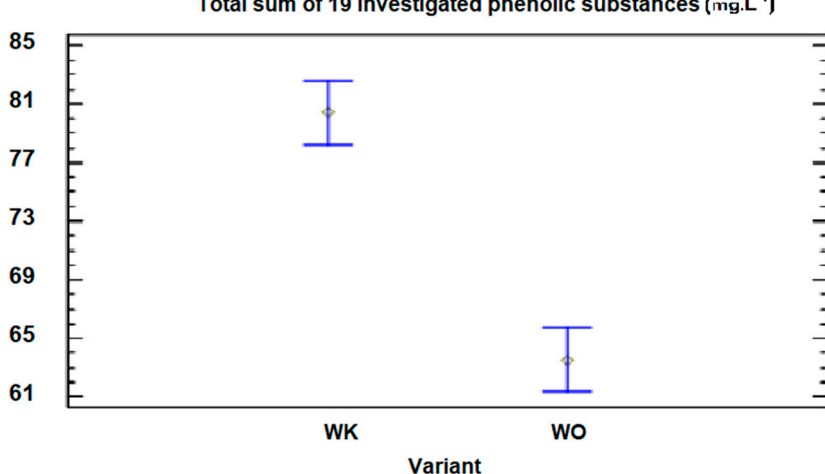

**Figure 2.** Graphic representation of the 95% confidence intervals of the content of 19 monitored phenolic substances for the tested averages from individual variants in 2015 and 2016 for the Welschriesling variety.

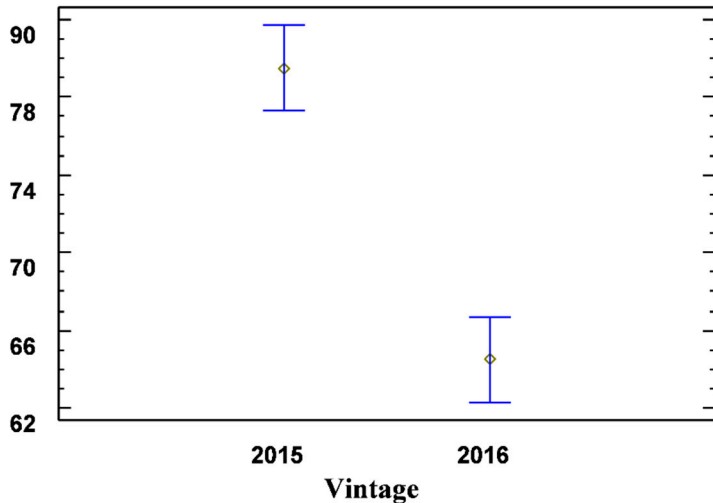

**Figure 3.** Graphic representation of the 95% confidence intervals of the content of 19 monitored phenolic substances for the tested averages in 2015 and 2016 for the Welschriesling variety.

For the variant produced by the reductive method (the control variant) in the year 2015, the total content was 88.37 mg·L$^{-1}$ (with standard deviations of ±1.25), and for the variant that was processed by targeted oxygenation, it was 68.63 mg·L$^{-1}$ (with standard deviations of ±0.85). This represents a reduction of 21.5% (Table 4). The important phenolic substances, which cause the oxidative processes in wine, include the total caffeic acid (containing the camphoric acid), and its content in the control variant of the W variety was 38.8 mg·L$^{-1}$ (with standard deviations of ±0.98). For the targeted oxygenated variant, it was 32.6 mg·L$^{-1}$ (with standard deviations of ±0.51). In [19], the effect of wine lying on yeast from 60 to 360 days was monitored to determine the hydroxycinnamic acid content (such as caffeic acid). Values from 23.40 to 30.33 mg·L$^{-1}$ were observed, while the time spent lying on the yeast did not affect the content of this group of the phenolic substances. However, a significant decrease in the content of the total phenolic substances was observed. For example, for the Chardonnay variety, the values were from 373.80 to 124.95 mg·L$^{-1}$. In our research, we found that the content of the monitored 19 phenolic

substances decreased by 12.8% for the Chardonnay variety, from 66.35 (the control variant) in 2016 to 57.86 mg·L$^{-1}$ (the variant processed by oxygenation).

The year 2016 was significantly different in the terms of the climatic conditions compared to the year 2015. The above-average total precipitation, the lower number of sunny days, and the cold autumn caused a lower ripeness of the grapes and lower concentrations of the constituents in the must, including phenolics. The total content of 19 monitored phenolic substances in the control variant of the Riesling variety was 72.03 mg·L$^{-1}$, and in the oxygenated variant, it was 56.96 mg·L$^{-1}$. Camphoric acid is the predominant hydroxycinnamic acid. According to [20], caftaric acid is significantly affected by location. The content of caftaric acid in the Chardonnay variety from the Sádek area was 37.11 mg·L$^{-1}$, and in the Perná site, it was 24.00 mg·L$^{-1}$. These values are comparable to our results for the same variety. We observed values from 33.92 to 47.23 mg·L$^{-1}$. In our research, camphoric acid was evaluated for a better overview; as with its esters and derivatives, it is the equivalent of the total caffeic acid. Unlike these authors, we did not find significant differences in the content of the phenolic substances within the localities, but we found them over the years. In [21], the following values of camphoric acid in the Chardonnay variety were reported: 191 mg·L$^{-1}$ in the freshly pressed must content, 86 mg·L$^{-1}$ in the free-flowing part, and 73 mg·L$^{-1}$ in the must after oxygenation (12 hours of musting). Moreover, in [22], it was determined that there were 13.43 mg·L$^{-1}$ and 25.29 mg·L$^{-1}$ of camphoric acid in the must and in the pressed fraction, respectively. The content of the flavonoid substances in the free flow was 0.66 mg·L$^{-1}$, and in the compressed fraction, it was up to 15.70 mg·L$^{-1}$.

The contents of substances of a non-flavonoid nature in white wine were found to be 164.5 to 245.5 mg·L$^{-1}$ [23]. Their contents decreased with the aging of wine. The nephlavonoids in red wine were found to be from 232 to 377 mg·L$^{-1}$. The content of the flavonoids in white wine was found, in the present study, to be in the range of 15 to 25 mg·L$^{-1}$, and in red wine, it was in the range of 100 to 200 mg·L$^{-1}$. In our research, we found 3.2 to 10.1 mg·L$^{-1}$ of flavonoids (as the catechin and epicatechin equivalent) in the individual variants. The content of flavonoids in each variant and year was always higher in the control variant than in the oxygenated variant (Tables 3 and 4). Other researchers have found up to 91 mg·L$^{-1}$ of flavonoids (as the catechin equivalent) in Sauvignon Blanc wine made from mechanized harvest grapes [24]. The authors did not find a negative effect of must adequately exposed to atmospheric oxygen on wine quality.

The content of polyphenols in Georgian white wines range from 1.330 to 2.430 mg·L$^{-1}$. In red wines, their content is from 2.898 to 4.416 mg·L$^{-1}$. Many Georgian wines have a high phenolic content, as white wines are also produced by the fermentation maceration of must [25]. The long-term aging of wine on fine yeast sludge eliminates the content of phenolic substances in wine. Their absorption occurs in such a way that the polysaccharides and the mannoproteins released from yeast sludge react with the phenolic substances [26].

In addition to the decrease in the content of total caffeic acid, we found a decrease in the content of gallic acid and its ethyl ester, 4-hydroxybenzoic acid, coumaric acid, and ferulic acid in the samples produced by the targeted oxygenation of must. The content of gallic acid in the Chardonnay variety in 2015 ranged from 1.85 to 2.12 mg·L$^{-1}$ (Table 3) [20]. This indicated values of 0.87 to 0.95 mg·L$^{-1}$ for wine of the same variety. In [27], the phenolic substances of white wines produced from the Narince grapes harvested from the different localities were studied. The contents of the Gallic acid of the must obtained from the grapes harvested from Erbaa and Emirseyit were 2.33 mg·L$^{-1}$ and 2.45 mg·L$^{-1}$, respectively. The contents of ferulic acid were 1.79 mg·L$^{-1}$ and 1.41 mg·L$^{-1}$, and the contents of caffeic acid were 1.0 mg·L$^{-1}$ and 0.69 mg·L$^{-1}$, respectively. The contents of gallic acid at the end of fermentation were 3.49 mg·L$^{-1}$ and 3.09 mg·L$^{-1}$, respectively. The contents of ferulic acid were 1.87 mg·L$^{-1}$ and 1.44 mg·L$^{-1}$, and the contents of caffeic acid were 3.10 mg·L$^{-1}$ and 2.82 mg·L$^{-1}$, respectively. Consequently, at the end of the clarification process, the content of total phenols decreased statistically significantly. The authors do not list the types of clarification substances used. The differences in the total phenolics of the must and the wines produced from the grapes harvested from two different localities were not found to be significant. These results do not agree

with the claims in [20] that phenolic substances can be good markers of terroir [28]. Differences in the content of the phenolic substances in must in different phases of must pressing were observed. At the beginning of the hydraulic press, it was determined to be 340 mg·L$^{-1}$ for the Welschriesling variety and 370 mg·L$^{-1}$ at the end of the total polyphenols. It is recommended to reduce their content in white wines using technological additives, e.g., gelatine [29]. In a similar experiment, which focused on the effect of different types of grape harvesting and processing techniques on the protein and the phenolic content of wine, they found that the free-run juice contains the lowest content of phenolic substances but the most proteins, as these are found mainly in the pulp of the berry. These facts must be considered in the technological processes of wine production, because at present, in order to eliminate phenolic substances, the methods of pressing whole grapes or separating the free flow are often applied. In these cases, the lower protein (colloidal) stability of the wine must be taken into consideration, thus determining the method of clarification and the dose of the clarifiers.

We performed sensory evaluations twice in 2015 and 2016, each time in February and November, in order to monitor the maturing potential of the wine. The wines produced by the reductive technology and the targeted oxygenation after maturing for 9 months received a comparable point, as assessed by sensory analysis. The differences in the evaluation of the 100 point scale were irregular and inconclusive and represented a maximum of two points. Such a point difference must be negligible in understanding the sensory quality of wine. In the variants processed by targeted oxygenation, the descriptors of minerality, fruit ripeness, and tertiary bouquet were highlighted after 9 months of wine maturation. The wines' color in the oxygenation variant ranged from green to yellow tones. The brown shades had already appeared in some samples of the control variant as a result of the oxidation of the phenolic substances in parallel with a loss of the Sulphur dioxide activity. The wines produced by the must oxygenation method have a more balanced and complex sensory profile.

The evaluation of the specific taste descriptors for the Welschriesling 2015 variant processed by the reductive method is shown in Figure 4. The evaluation was conducted on 30 November 2016. The descriptors were evaluated by a five point scale (0–5). The positively evaluated attributes from the first sensorial evaluation of the young wine were lost as early as after 9 months of ripening. The oxygenation and the nuances of the yeast decomposition in the wine removed the taste of ripe apple. The phenolic substances, after losing the effectiveness of free SO$_2$, triggered the oxygenation processes. The final evaluation was 80.6 points.

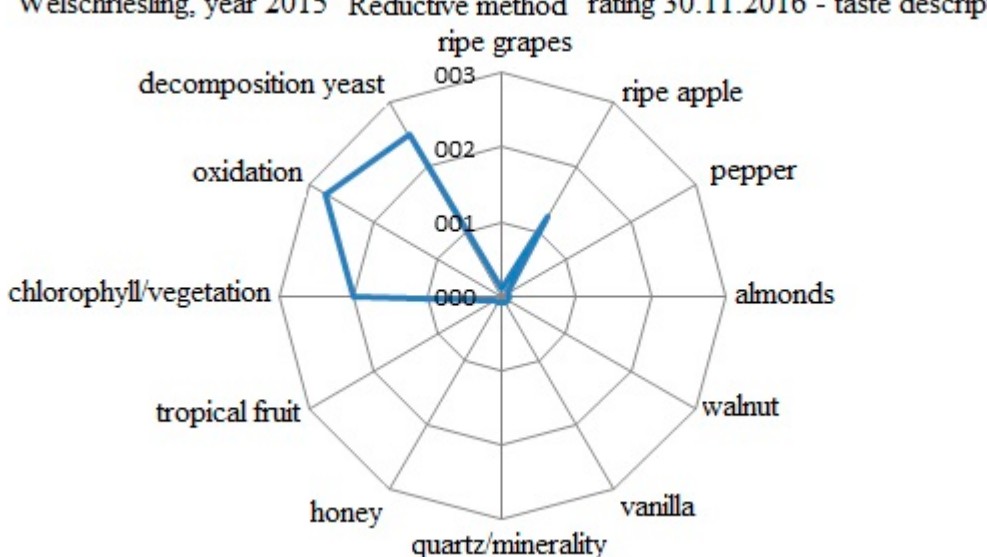

**Figure 4.** Evaluation of the specific taste descriptors for the W 2015 variant processed by the reductive method.

The evaluation of the specific taste descriptors for the Welschriesling 2015 variant produced by targeted oxygenation is shown in Figure 5. The evaluation was conducted on 30 November 2016. The descriptors were evaluated by a five point scale (0−5). The tastes of ripe apple and minerality/quartz were dominant. It was found that the nuances of the ripe grapes and the walnuts were positive. The variant produced by targeted oxygenation, evaluated 12 months after grape harvesting, was 3.7 points higher than that produced by the reductive method evaluated at the same time. The tone of tropical fruit was substituted by minerality, which is, from the point of view of archival potential, really positive. The final evaluation was 84.3 points.

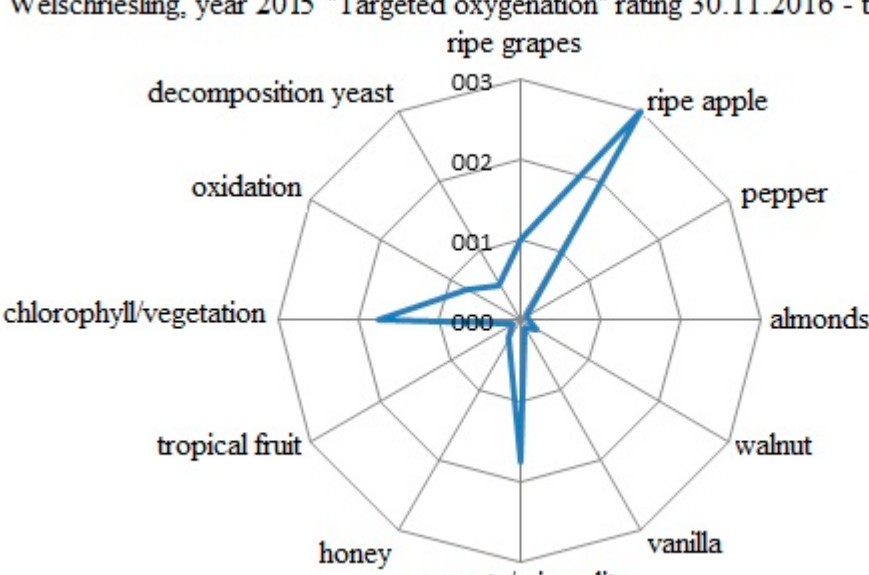

**Figure 5.** Evaluation of the specific taste descriptors for the W 2015 variant processed by targeted oxygenation.

The evaluation of the specific taste descriptors for the Welschriesling 2016 variant processed by targeted oxygenation is shown in Figure 6. The evaluation was conducted on 23 November 2017. The descriptors were evaluated by a five point scale (0−5). Even after 12 months of grape harvesting, reductive plant tones and succulent ripe apple were predominant. The wine was fresh and vivid, and despite the fact that its result potential was on the top, it did not become a typical grape variety. The final evaluation was 84.6 points.

The evaluation of the specific taste descriptors for the Welschriesling 2016 variant produced by targeted oxygenation is shown in Figure 7. The evaluation was held on 23 October 2017. The descriptors were evaluated by a five point scale (0−5). In the wine, succulent ripe apple predominated, which was considered to be the typical expression of the growing year, and organic acids followed from it. The dominance of ripe apple appropriately refills the variety of the typical tones of walnut, ripe grape, minerality/quartz, and copper. The green tones were appropriately integrated into the wine texture, but the particular components of the taste attributes were still not at the required harmony. The total impression of the wine indicated a potential for further ripening and development of the sensory profile. The final evaluation was 84.2 points.

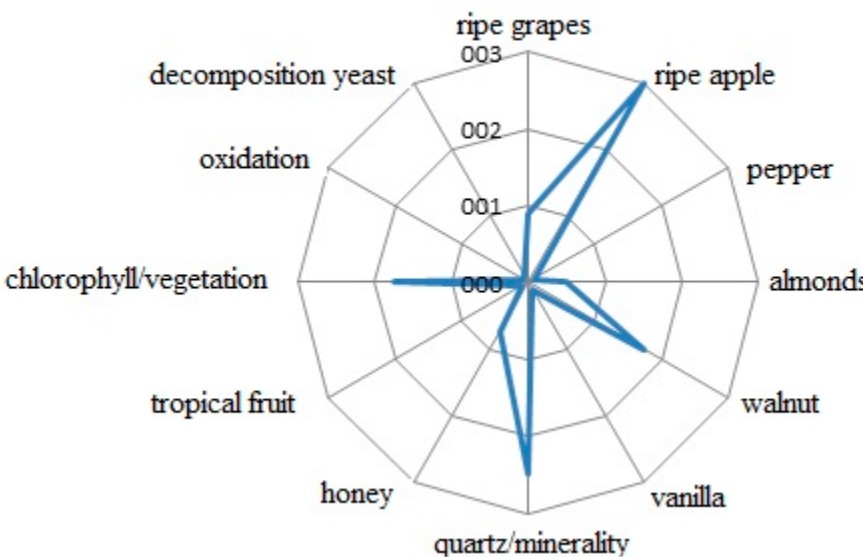

**Figure 6.** Evaluation of the specific taste descriptors for the W 2016 variant processed by the reductive method.

Welschriesling, year 2016 "Targeted oxygenation" rating 23.10.2017 - taste descriptors

**Figure 7.** Evaluation of the specific taste descriptors for the W 2016 variant processed by targeted oxygenation.

## 4. Conclusions

We achieved positive results by applying targeted oxygenation technology. In two of the three monitored varieties, we achieved a statistically significant reduction in the content of phenolic substances. If targeted oxygenation is carried out simultaneously with pre-fermentation maceration of must, wines are richer in aromatic substances, they are more powerful, and there is less need for sulphating them. It is possible to naturally reduce the additives by at least one dose of $SO_2$ (30 mg·$L^{-1}$). The content of phenolic substances in must and wine can also be reduced using technological auxiliaries. However, the oxygenation method makes this process gentler, more natural, and finally, more economical.

Excess oxygen is consumed by yeast during fermentation to form esters and the higher fatty acids needed for their construction. Alcoholic fermentation is healthier and smoother in oxygenated must.

Targeted oxygenation can also be a suitable technology to avoid uniformity of wines, which often arises from the use of strictly reductive technology. The most important result of the sensory evaluation is that the technology for the targeted oxygenation of wine does not cause any sensory damage.

We consider the most suitable way of applying the experiment in practice to oxidize must for 8 hours and oxidize must in sludge for 8 hours at a low temperature of up to 15 °C. We recommend the targeted oxygenation technology primarily in the production of wines from grapes with a sugar content of max. 230 g·L$^{-1}$. We will continue the present research, focusing on a more accurate control of the process and the method of oxygenation and examining the influence of exposure time, temperature, and volume of incoming oxygen in order to optimize the work procedures.

The highest content of the substances of a neoflavanoid character was found in the Welschriesling variant—the controlled variant—in the year 2015, at 80.69 mg·L$^{-1}$, and in the oxygenated variant, their content was 59.42 mg·L$^{-1}$. The content of flavonoids in the oxygenated variant was 7.25 mg·L$^{-1}$, and in the controlled variant, it was 8.27 mg·L$^{-1}$.

**Author Contributions:** Conceptualization, Š.A.; Data curation, J.P. and Š.A.; Investigation, Š.A. and J.J.; Methodology, Š.A. and P.C.; Writing—original draft, J.P. and Š.A.; Writing—review and editing, J.P. and Ľ.J. All authors have read and agreed to the published version of the manuscript.

**Funding:** "This research was funded by the Slovak Research and Development Agency (APVV), grant number 16-0278". "This work was supported by AgroBioTech Research Centre, built in accordance with the project Building, "AgroBioTech" Research Centre, ITMS 26220220180".

**Conflicts of Interest:** The authors declare no conflict of interest. The funders had no role in the design of the study; in the collection, analyses, or interpretation of data; in the writing of the manuscript; or in the decision to publish the results.

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
