# Peer review of "The Use of a Targeted Must Oxygenation Method in the Process of Developing the Archival Potential of Natural Wine"

_applsci, doi:10.3390/app10144810_

Round 1
Reviewer 1 Report
The article entitled "Targeted must oxygenation method use in the process for developing the natural wine archival potential" aims on the comparative study on the qualitative characteristics of wines through the application of two pre-fermentation methods, the reductive (conventional) method and a method of directed oxidation in the must.
From an enological point of view, it is a very interesting study, as it could result in improvements and technology transfer in the wineries.
Suggestions:
In this sense, these are suggestions to be taken into account to improve the quality of the text of the manuscript.
Abstract:
- Line 17:Authors should change in the text the word "biological" for "vegetable". It would be more correct, since they develop a research focused on the physical-chemical compounds of the musts and mainly of the wines. Instead of carrying out a study focused on the plant (vegetative development, physiology, stress, etc.), in which it would be more correct to use the expression "biological material".
1. Introduction
- Lines 42-43: Although it is well known to experts in oenology and winemaking, authors should include some citations to support this statement.
- Lines 44-47:Authors should re-write these sentences, better connecting the ideas they give and providing some bibliographical citation to support the data indicated.
- Lines 49-52: The authors indicate that several authors consider two big groups for phenolic substances, however no citation is given in the text to support this statement.
- Line 56: The citation number [7] is not well placed. It should be relocated correctly.
- Lines 67-73: All this information could make more sense if it were placed just before line 63 (the aim of the research.....).
- Lines 74-78:It seems more correct to place this information in section 2.Materials and Methods.
2. Materials and Methods
A sub-section could be included specifying the characteristics related to the plant material used.
Firstly, the authors should indicate in the text the correct name of the varieties with which they have carried out the research work. Even if the area where the grapes studied in this work were grown is then specified.
According to the Vitis International Variety Catalogue
-Italian Riesling is a synonym for the variety Welschriesling (prime name). A variety of Italian origin, of the species Vitis vinifera linné subsp. vinifera, with white berries.
-Rheinriesling is a synonym for Riesling weiss. A variety of German origin, of the species Vitis vinifera linné subsp. vinifera, with white berries.
- Lines 84-86: Authors should provide supplementary data on the climatic conditions of the area where the grapes were grown, on the date indicated in the paper. This could justify the climatic behaviour indicated in the text in lines 91-111.
- Lines 116-119: Authors should specify the doses of pectolytic enzymes and sulphur dioxide that were added to the paste, as well as the pressure used in the pneumatic press and the dose of bentonite added before the decantation of the must.
- Lines 120-127: The authors should include the doses of enzymes indicated by the manufacturer, as well as the brand and manufacturer of these enzymes. In addition, they should indicate what type of pure yeast culture was used and the dosage.
- Line 127: Add "C" to the temperature to correctly express the degrees Celsius.
- Lines 132-135: Authors should indicate the conditions of the chromatographic method used in the analysis of phenolic compounds. They should also specify whether they have made a calibration line with standards for each of the compounds studied.
- Line 137: Authors should include in the text the bibliographic reference of the method used for the sensory analysis of the wines.
- Line 142: Authors should specify in the text which are the specific attributes evaluated in the wines.
3. Results
The authors make a good discussion of the evolution of the phenolic compounds in the different experiments.
Tables 1, 2, 3 and 4: Authors should include in the tables the values of the standard deviations, as well as indicate the number of replicates analyzed for each of the parameters studied.
Have the authors checked whether the values follow a normal distribution before performing the statistical analysis?
- Line 231: Leave a space between "from" and "232".Write in superscript "-1" in mg.L
- Line 235: Citation [18] does not appear to be properly placed. Replace it.
- Line 251: Write in superscript "-1" in 1.41 mg.L
- Lines 271-281: Authors could include a figure to facilitate the understanding of the results obtained in the sensory analysis of wines. In the same figure, it could be indicated which of these results are the most significant due to the methods studied.
4. Conclusions
Authors should correct the numbering of the " Conclusions " heading, it corresponds to a 4.
I would like to congratulate the authors for the work done and the detail in which the different analyses were carried out.
Author Response
Dear Reviewer,
we feel great thanks for your professional review work on our article, your suggestion really means a lot to us. We supplemented our results and edited our article extensively. According to Reviewer suggestions, we have made corrections to our previous manuscript. Detailed corrections are given in the attached document.

Reviewer 2 Report
The manuscript investigates the influence of oxygenation during winemaking on phenolic concentration in wine, which is of interest to the field. However the manuscript requires substantially greater detail before publication. In particular, more details are requested for the methods section, specifically how wines were made, volumes of each batch and the number of wine batch replicates per variable. Currently it appears as though each variable was only produced as a single batch instead of in triplicate. If this is the case, no conclusions can be drawn about the impact of oxygen exposure during winemaking on phenolics, and the paper cannot be published in a scientific journal. The results included in this manuscript may therefore be better used as the basis for further research. If the wines were made in triplicate, this needs to be specified in the methods and standard deviations provided for each of the phenolic concentrations for each variable in the results. The sensory methods and results also need to be shown in greater detail including references and results tables and/or figures. The manuscript would also benefit from editing by a native English speaker to correct the numerous inaccuracies of expression.
More details are provided below.
Introduction:
Line 40: Change “compounds” to aggregates
Line 43: Include a reference
Line 45: include a reference. Change “It is caused” to “Phenolic concentration is influenced” for clarity.
Line 47: Correct the term “stum” and include a reference
Line 54: Re-write this sentence for accuracy and clarity. Sugars and proteins do not contribute to wine colour.
Methods:
Tabulate the climate information for easy reference. Include additional information in the introduction about the influence of climate on phenolic development in grapes for context.
Line 113: Correct the term “second and third decade of September”. Perhaps “week” is more accurate than “decade”?
Change “mush” to “must” throughout the manuscript.
Use subheadings to separate methods used for winemaking, chemical analyses and sensory analyses.
Include more specific winemaking details, particularly how and when oxygen was introduced during winemaking. Include information about replicates for the winemaking. Add details about the sensory analyses such as the number of sensory panellists, provide a reference for the IUO method.
Results:
Table 3: Include standard deviations for phenolic concentrations. Were wine batches produced in triplicate? Line 284 indicates that the reduction was “statistically significant” but this is not supported by the presented data. This needs to be included before the manuscript can be accepted for publication in a scientific journal.
Tables 3 and 4: Suggest changing the configuration with variables across the top and measured phenolic species down the left hand side for easier comparison.
Lines 199-211: Without standard deviations from wine batch replicates it is impossible to assess if the reduction in phenolics induced by oxygenation is significant.
Line 271-281: Show the sensory results as tables or PCA plots.
Author Response

(The authors gave the same response as above.)

Reviewer 3 Report
The article is confusedly written and shows many flaws.
First of it is exposed in a twisted and unclear way. In the materials and methods section, the authors insert a description of the weather trend of the two years considered that could be summarized in a table, lightening the section.
The number of repetitions used in analytical determinations is completely lacking, as well as the standard deviation in the tables. The determination of the reducing sugars is redundant if the glucose and fructose data are exposed.
The sensory profile of the wines obtained is not illustrated. In my opinion, to highlight the influence of the technique studied, this is more important than the scoring evaluation of the International Union of Oenologist
Author Response

(The authors gave the same response as above.)

Round 2
Reviewer 1 Report
The authors have faithfully followed each of the suggestions and the manuscript has exponentially increased in quality.
I would like to congratulate all the authors for their work.
Author Response
Thank you
You have helped us a lot in improving the quality of our article.
We submitted the article for review by authoring services for editing the MDPI in English. The latest version of the article is after publisher language modifications.
Thank you and have a nice day
Reviewer 2 Report
The revised manuscript is substantially improved with more details of the methodology used and better presentation and discussion of results.
Some minor corrections to spelling and grammar are required throughout the manuscript but otherwise is suitable for publication.
Author Response

(The authors gave the same response as above.)

Reviewer 3 Report
I have noted with pleasure that the comments have been accepted.
The paper has improved significantly, but there are still some inaccuracies: please check the exact typing of some words (e.g. row 90 Chardoney instead Chardonnay, row 198 ...ethnology... instead oenology, etc....)
The sensory profiles can be combined in two graphs, one for each vintage, so that the effect of the need can be directly compared.
Author Response
Thank you
You have helped us a lot in improving the quality of our article.
Response to Reviewer 3 Comments
- row 90: Chardoney Chardonnay became an integral part of our assortment in the wine production
- row 198: to The international ethnology oenology union [18].
- Your latest chart change recommendations are included in the latest version of the article.
We submitted the article for review by authoring services for editing the MDPI in English. The latest version of the article is after publisher language modifications.
Thank you and have a nice day